# Peak Power Output Is Similarly Recovered After Three- and Five-Days’ Rest Following Sprint Interval Training in Young and Older Adults

**DOI:** 10.3390/sports7040094

**Published:** 2019-04-25

**Authors:** Zerbu Yasar, Susan Dewhurst, Lawrence D. Hayes

**Affiliations:** 1Active Ageing Research Group, Department of Medical and Sport Sciences, University of Cumbria, Lancaster LA1 3JD, UK; Lawrence.hayes@cumbria.ac.uk; 2Department of Sport and Physical Activity, Bournemouth University, Poole BH12 5BB, UK; sdewhurst@bournemouth.ac.uk

**Keywords:** high-intensity interval training, maximal, older adults, peak power output, recovery, sprint interval training

## Abstract

(1) Background: High-intensity interval training (HIIT) exerts effects indicative of improved health in young and older populations. However, prescribing analogous training programmes is inappropriate, as recovery from HIIT is different between young and older individuals. Sprint interval training (SIT) is a derivative of HIIT but with shorter, maximal effort intervals. Prior to prescribing this mode of training, it is imperative to understand the recovery period to prevent residual fatigue affecting subsequent adaptations. (2) Methods: Nine older (6M/3F; mean age of 70 ± 8 years) and nine young (6M/3F; mean age of 24 ± 3 years) participants performed a baseline peak power output (PPO) test. Subsequently, two SIT sessions consisting of three repetitions of 20 s ‘all-out’ stationary cycling bouts interspersed by 3 minutes of self-paced recovery were performed. SIT sessions were followed by 3 days’ rest and 5 days’ rest on two separate occasions, in a randomised crossover design. PPO was measured again to determine whether recovery had been achieved after 3 days or after 5 days. (3) Results: Two-way repeated measure (age (older, young) × 3 time (baseline, 3 days, 5 days)) ANOVA revealed a large effect of age (*p* = 0.002, *n^2^_p_* = 0.460), with older participants having a lower PPO compared to young participants. A small effect of time (*p* = 0.702, *n^2^_p_* = 0.022), and a medium interaction between age and time (*p* = 0.098, *n^2^_p_* = 0.135) was observed. (4) Conclusions: This study demonstrates both young and older adults recover PPO following 3 and 5 days’ rest. As such, both groups could undertake SIT following three days of rest, without a reduction in PPO.

## 1. Introduction

High-intensity interval training (HIIT) is characterised by exercise above 80% of maximum heart rate interspersed with lower-intensity recovery phases [1]. Sprint interval training (SIT), a derivative of HIIT, is characterised by maximal exertion, sustained for 30 s or less [2]. HIIT has gained popularity due to improvements in fitness comparable to moderate-intensity continuous training (MICT) [3,4]. Similarly, SIT has produced comparable adaptations to MICT in young adults [2,5], including increased aerobic function, a key a determinant of long-term mortality [6]. With the reported increased enjoyment [7,8], and time efficiency [9], this supports promotion of HIIT or SIT over MICT, by removing potential barriers to exercise adherence.

A limitation of HIIT is that it generally requires intensity-based calculations [1,10,11], which are not required in the use of ‘all-out’ SIT protocols. Previous HIIT research has shown effects indicative of improved health with three HIIT sessions per week, in a group with a mean age of 70 years [12], and groups with a mean age of 63 years [13], and both 60–69 years and 70+ years [14]. However, in other studies, similar benefits have been seen with a single session every 5 days, with a mean age range of approximately 60 to 63 years [15,16,17,18]. HIIT every five days has been found to increase peak power output (PPO) in older individuals with a mean age of ~61 years [18,19]. PPO is a physiological measure of paramount importance to the older individual due to its importance in physical functioning [20], which elevates the profile of HIIT as a training method to target improvement in power output. However, Herbert et al. [21] observed a delayed recovery of PPO in older participants with a mean age of 63 years. The exercise protocol utilised was a HIIT session comprising of 6 × 30 s intervals working at 50% of PPO, interspersed with 3-minutes active recovery phases. Therefore, some caution is required when prescribing HIIT to older adults. Moreover, it is possible that a delayed recovery in older individuals may transcend between HIIT and its derivative form of ‘all-out’ SIT, which employs a higher intensity.

Delayed recovery seen in older adults compared to their younger counterparts could be attributed to several biological processes [22]. The driver of the delayed response may be attributed to an age-associated reduction in mitochondrial function [23]. Consequently, a delayed recovery response may be initiated following exercise due to dysregulated reactive oxygen species production and regulation in ageing skeletal muscle [24]. For a more detailed review of the mechanisms of skeletal muscle ageing, we suggest a previous review [25]. The primary effects of ageing on skeletal muscle are sarcopenia, defined as the loss of muscle, and dynapenia, defined as the loss of force production [26]. Although the reduction of muscle strength and mass is a key determinant of physical function [27], muscle power may be a more important determinant of functional capacity [20]. Importantly, this decline in physical function is associated with increased incidence of physical disability, loss of independence, and increased mortality [27].

SIT has demonstrated positive power adaptations in young cohorts [28,29,30,31,32]. This increase in power development appears to be maintained at low-volume SIT training loads of 4–6 repetitions of 10 s maximal sprints when compared to longer 30 s sprints in recreationally active young adults [33]. Additionally, improvements have been observed in aerobic fitness with the prescription of SIT in young cohorts with Wingate SIT protocols consisting of 4–6 repetitions of 30 s maximal efforts [34,35]. Interestingly however, even at significantly reduced training volumes (2 × 20 s sprints), SIT was effective at increasing aerobic function in young populations [36,37]. At present, there is a paucity of data concerning SIT in older adults. 

The previously discussed literature justifies the investigation of a low-volume SIT protocol as an intervention to increase overall physical function in older adults. However, before the adaptations to a 3 × 20 s ‘all-out’ SIT exercise training programme can be explored in older adults, it is imperative to know the duration of rest sufficient for post-training PPO recovery. A previous review has discussed literature on HIIT intersession recovery, concluding optimal recovery to be approximately 48 h following HIIT [11]. However, research discussed in this review concerned young and athletic populations, who likely recover faster due to age-related biological factors [22,23,24]. Therefore, the present study examined recovery timeframes utilised by Herbert et al. [21]. This is important to avoid maladaptation, but also to avoid a period of reduced muscle power which, as previously discussed, would result in diminished functional capacity. Therefore, the aim of the present study was to investigate PPO after 3 days’ and 5 days’ recovery following a cycling SIT session in young and older participants. We hypothesised *a priori* that for PPO restoration, older participants would require 5 days’ recovery, and young individuals would recover after 3 days.

## 2. Materials and Methods

### 2.1. Participants

This study was carried out in accordance with the Declaration of Helsinki and approved by the University of Cumbria Research Ethics Committee (Reference code: 16/74). Written informed consent was obtained from all participants prior to study commencement. A Physical Activity Readiness Questionnaire (PAR-Q) and American College of Sports Medicine (ACSM) pre-exercise participation screening were completed [38]. Participants were habitually physically active, exercising at least twice a week, totalling at least 150 minutes of moderate exercise. Nine older (6M/3F; mean age of 70 ± 8 years, height of 174 ± 9 cm, mass of 70 ± 10 kg) and nine young (6M/3F; mean age of 24 ± 3 years, height of 174 ± 9 cm, mass of 73 ± 7 kg) individuals participated. Abstention from alcohol, caffeine, and exhaustive exercise was required for 24 hours prior to testing sessions.

On the first visit, a baseline PPO assessment was completed. Seven to ten days later, participants performed a SIT session. This exercise was followed by 3 days’ rest or 5 days’ rest in a randomised crossover design, after which they returned to complete a second PPO measure. Subsequently, participants returned 7–10 days later to complete the other arm of the study (i.e., SIT session with 3 days’ or 5 days’ recovery).

### 2.2. Session 1: Baseline Peak Power Output 

Following measurement of stature and body mass, participants mounted the cycle ergometer, which was set up according to manufacturer's guidelines (Wattbike Pro, Wattbike Ltd, Nottingham, UK). Subsequently, participants warmed up for 6 minutes at approximately 70 W, interspersed with three ~2 s maximal sprints with an air brake resistance of 8 and a magnetic resistance of 1. Following 5 minutes of passive recovery, participants performed a 6 s Herbert test [39], which involved a maximal sprint from a stationary start, with the air brake set to 10 and magnetic resistance set to 1. Power output was calculated each second for the duration of the test, and PPO was considered as the highest value over 1 s.

### 2.3. Session 2 and 4: Sprint Interval Training and Peak Power Assessment

As above, participants warmed up for 6 minutes at approximately 70 W, interspersed with three ~2 s maximal sprints with an air brake resistance of 8 and a magnetic resistance of 1. Following 5 minutes of passive recovery, participants remounted the ergometer with the air brake resistance set to 3 and magnetic resistance set to 1. Participants completed 3 × 20 s maximal sprints, interspersed with 3 minutes of active recovery, with strong verbal encouragement during each sprint (Figure 1). A summary of work performed by participants is displayed in Table 1. Upon completion of the final maximal effort interval, a 5-minutes self-paced cool down was performed. Following either 3 or 5 days’ recovery, a Herbert 6 s test [39] was repeated to determine PPO.

### 2.4. Statistical Analysis 

Statistics were processed using SPSS version 23.0 (IBM). Following a Shapiro–Wilk’s test of normality and Levene's test for homogeneity of variance, a two-way repeated measures ANOVA (age (young vs. older) × recovery time (baseline, 3 days’ rest, 5 days’ rest)) was conducted. Alpha level was set a priori at *p* < 0.05. Partial eta squared (*n­_­­_^2^_p_*) was used as a measure of main effect, defined as small 0.02, medium 0.13, and large 0.26. Cohen's *d* was calculated for pairwise comparisons Additionally, an independent samples *t*-test was conducted to compare weekly mean habitual physical activity at above moderate intensity between older and young participants. Effect size was determined using Cohen’s *d*, defined as small 0.1, medium 0.3, and large 0.5. Data are presented as means ± standard deviation (SD). 

## 3. Results

A large age effect (*p =* 0.002, *n^2^_p_* = 0.460), small time effect (*p =* 0.702, *n­_­­_^2^_p_* = 0.022), and medium interaction effect (*p* = 0.098, *n­_­­_^2^_p_ =* 0.135) was present for PPO. Younger participants produced greater PPO than older participants (Figure 2). Young PPO for baseline, 3 days’ rest, and 5 days’ rest was 942 ± 274 W, 921 ± 260 W, and 913 ± 258 W, respectively (Cohen's *d* < 0.11 for all pairwise comparisons). Older PPO for baseline, 3 days’ rest, and 5 days’ rest was 543 ± 151 W, 561 ± 152 W, and 555 ± 152 W, respectively (Cohen's *d* < 0.12 for all pairwise comparisons). Weekly mean habitual physical activity revealed a large effect between older (417 ± 313 minutes) and young participants (310 ± 65 minutes; *t* = (8.69) 1.01, *p* = 0.342, *d* = 0.68); equal variances were not assumed (*p* = 0.14).

## 4. Discussion

The main finding of the present study was that young and older individuals recover PPO from a single SIT session after 3 days’ rest. To our knowledge, this is the first study which has investigated recovery following SIT in older adults, and data presented here suggest that recreationally active older adults can include SIT into their physical activity programmes with 3 days’ rest, without detriments to PPO.

Current physical activity guidelines for older people suggest that at least 150 minutes of moderate, or 75 minutes of vigorous aerobic exercise should be accumulated weekly in at least 30- or 10-minutes bouts, respectively [40]. Additionally, Chodzko-Zajko et al. [40] suggested a resistance training frequency of twice per week. Currently, however, there is no comparable consensus on HIIT or SIT for older adults. Some evidence has emerged in attempting to provide prescriptive guidelines for HIIT by Herbert et al. [21]. This research demonstrated a delayed PPO recovery from HIIT in older males compared to young males (5 days versus 3 days respectively). Data from the present investigation differ from those of the HIIT-based recovery study by Herbert et al. [21] in that we have demonstrated PPO recovery from SIT after 3 days. This suggests that PPO recovery from HIIT and SIT are different in older adults. 

The intensity of the protocol used in the present study was ‘all out’ or maximal power output, as opposed to the 50% of peak power output (~120% peak oxygen uptake), maintained for 30 s used by Herbert et al. [21]. Given that a higher intensity was utilised in the present study, intensity is unlikely to be the determining factor in PPO recovery duration. The most obvious difference is the greater volume and duration of the exercise protocol employed by Herbert et al. [21]. For instance, the present study used three 20 s maximal intervals, rather than six 30 s intervals at a sustained 50% of maximal effort (i.e., 60 s total work vs. 180 s total work). Previous observations have noted exercise increases the production of reactive oxygen species [41]. Mechanistic investigations suggest that reactive oxygen species are produced as a by-product of mitochondrial respiration, and reactive oxygen species production is positively associated with oxidative phosphorylation [42]. Additionally, previous research has noted the lower overall energy demand of low-volume SIT protocols compared to typical HIIT protocols, even with consideration to the higher intensities used in SIT [1]. This suggests the possibility that the HIIT protocol employed by Herbert et al. [21] was more productive of reactive oxygen species in comparison with the 3 × 20 s SIT protocol used in the present study.

Excessive production of reactive oxygen species has been implicated in deleterious effects via inflammatory pathways to muscle function and performance [43]. Although reactive oxygen species are facilitative ­­­of physiological adaptations, it is theorised that there is an optimal reactive oxygen species production threshold, influenced by exercise intensity and/or duration, which may be altered with training status [44] and age [24]. Given that the participants were of a similar training status and age in both the present study and the study conducted by Herbert et al. [21], we tentatively speculate that the training stimulus provided by a 3 × 20 s SIT protocol, as used in the present study, may be more appropriate for reactive oxygen species regulation, as opposed to the protocol used by Herbert et al. [21]. However, this speculation requires further robust mechanistic evaluation within older age groups comparing HIIT and SIT protocols. Furthermore, it is noteworthy to mention that regular exercise has demonstrably improved ROS regulation [44]. Therefore, it is probable that ROS regulation would adapt with exercise habituation. 

Strength training studies in older adults demonstrate two or three sessions per week are optimal to facilitate adaptions [45,46,47]. Similarly, recent evidence suggests that aerobic training adaptations are optimised at a frequency of 3 to 4 times a week in older adults [48]. Strength training and aerobic training have been categorised as opposing ends of an exercise continuum [49]. However, prescribing from guidelines pertaining to either strength or aerobic training is evidently not appropriate when considering the delayed recovery time associated with HIIT [21]. 

Higher-frequency HIIT in older adults with a mean age of 63 years performed thrice weekly has increased peak oxygen uptake by 3% and 7% for women and men, respectively, over 6 weeks [13], approximately 11% for both 60 to 69 and 70 and above age groups over 8 weeks [14], and approximately 16% over 12 weeks [12]. However, a similar magnitude of improvement was observed, approximately 11% and 8% for previously sedentary and lifetime exercisers with a mean age range of 60–63 years, respectively, utilising lower-frequency HIIT performed once every 5 days for 6 weeks [15]. Importantly, only studies employing lower-frequency HIIT in older individuals have recorded PPO, with increases of approximately 17% in previously sedentary individuals with a mean age of approximately 62 [18], and 8% in master athletes with a mean age of approximately 60 years [19]. These data suggest lower-frequency HIIT may optimise aerobic improvements whilst increasing power in older individuals. Although strength and power are different variables, it is noteworthy to mention that the study by Robinson et al. [12] did not observe any increases to leg press strength following high-frequency HIIT training in older adults with a mean age of approximately 70 years. 

Previous research by Adamson et al. [50] demonstrated that repeated (6–10 repetitions) ×6 s sprints increased power by ~13% in an older cohort with a mean age of approximately 66 years, at a frequency of twice per week over a 10-week training intervention. This suggests that neurological adaptations are likely to be well targeted by shorter sprints. However, longer durations of 20 s are associated with increased metabolic stress, which is associated with increased mitochondrial biogenic messenger ribonucleic acid (mRNA) responses when compared to a work matched protocol consisting of shorter 5 s sprints [51]. This suggests that longer sprints are better optimised to increase aerobic function, which decreases overall mortality risk [6]. Due to the divergent stimulus provided by shorter sprints, previous reviews on the topic have justifiably differentiated this type of training as repeated sprint training (RST) [10]. Therefore, at present, we are unaware of any research regarding SIT in older individuals. 

We acknowledge the present study is not without limitations. For example, the use of recreationally active older adults was used in the current investigation, which does not permit application of these results to sedentary older adults. However, this recruitment strategy was necessary to ensure safe participation of older participants during maximal exercise [38]. Yet, HIIT has been used effectively in rehabilitation programmes for clinical populations in respiratory [52] and cardiac [53] pathologies, therefore demonstrating efficacy in ‘higher-risk’ cohorts. Importantly, the current findings may not translate to different modes of exercise, e.g., running, due to the associated increase in eccentric loading [54], which may increase recovery duration from exercise. Therefore, an investigation into recovery from different formats of sprint interval training is justified, with a particular focus on eccentric vs. concentric exercise load. Furthermore, the use of mixed gender sampling decreases the homogeneity of the groups included in the study.

In conclusion, the results of this study suggest that PPO recovery is similar between older and young adults after 3 days’ rest following SIT. These data permit SIT prescription in older adults, in the indicative knowledge that recreationally active individuals will be recovered after 3 days’ rest. We believe these data can guide prescription of SIT in healthy and active older individuals who may perform SIT following 3 days’ rest. As a strength of SIT over HIIT is that prescription is uncomplicated, future research may consider ecologically applicable modes of SIT in older adults, and whether SIT is a viable intervention to improve physical functioning.

## Figures and Tables

**Figure 1 sports-07-00094-f001:**
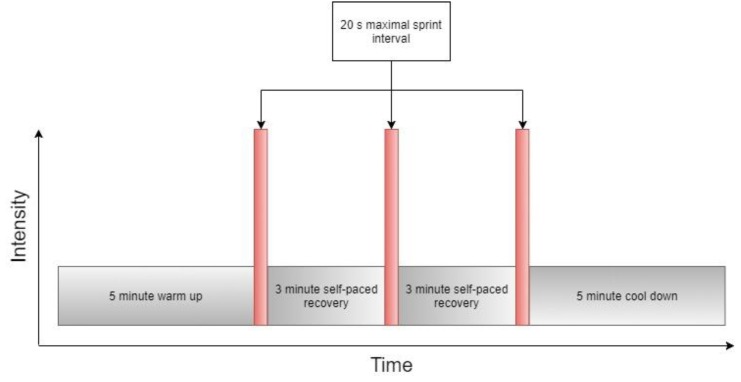
Schematic representation of sprint interval training protocol.

**Figure 2 sports-07-00094-f002:**
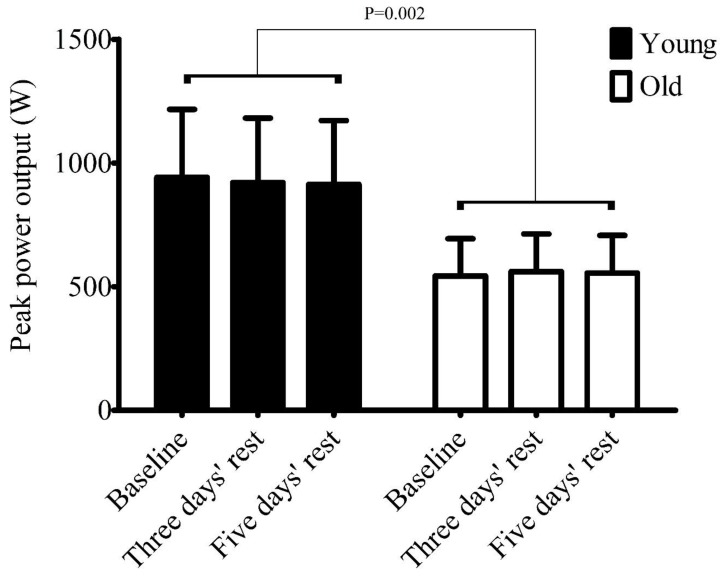
Peak power output (PPO) in young and older participants at baseline, after 3 days’ rest, and 5 days’ rest following sprint interval training (SIT). The alpha value of *p* = 0.002 indicates a significant difference between older and younger participants.

**Table 1 sports-07-00094-t001:** Amalgamated peak and mean power completed by older and younger participants during the sprint interval training session. Data are reported as mean and standard deviation (SD).

	Peak Power (W)			Mean Power (W)		
Group	Sprint 1	Sprint 2	Sprint 3	Sprint 1	Sprint 2	Sprint 3
Older	541 ± 135	528 ± 139	498 ± 146	402 ± 93	384 ± 93	362 ± 88
Younger	897 ± 246	828 ± 219	788 ± 215	579 ± 139	513 ± 148	473 ± 156

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
