# Peer review of "Peak Power Output Is Similarly Recovered After Three- and Five-Days’ Rest Following Sprint Interval Training in Young and Older Adults"

_sports, 2019, doi:10.3390/sports7040094_

Round 1

Reviewer 1 Report

The present paper examined the time course of recovery from SIT in healthy younger and older males. The paper presents some novel data, and I just have a few comments for the authors to consider.

Firstly, in the Introduction, please qualify "older" individuals. Is this >50, >60, >70 years? This will provide better context for the reader.

On page 2, line 49-50, when discussing the Herbert study, please explain how "delayed recovery" was defined or quantified.

Lines 78-81: Why were 3 and 5 days chosen? To my knowledge, most studies utilizing HIIT/SIT as a training methodology use ~48 hours or 2 days between training sessions. A stronger rationale is needed here.

Methods: Were participants given any pre-testing standardization instructions regarding physical activity and/or diet? These are potentially important confounders that should be addressed.

Results: While I understand the primary outcome was Peak Power, it would also be interesting to present data for mean power, minimum power, and/or fatigue index if these data are available. I feel this would strengthen the paper's primary outcomes.

Discussion: As with the introduction, I recommend quantifying the ages of other studies' participants so that it eases interpretation for the reader. Obviously we would likely expect a study with an average age of 50 to have different outcomes than a study with an average age of 70.

Thank you for the opportunity to review the paper.

Author Response

Reviewer 1 comments, and author responses.

The present paper examined the time course of recovery from SIT in healthy younger and older males. The paper presents some novel data, and I just have a few comments for the authors to consider.

Point 1: Firstly, in the Introduction, please qualify "older" individuals. Is this >50, >60, >70 years? This will provide better context for the reader.

Response 1: The qualification of old by the WHO is currently 60 years +. However, there are sub-classifications within this classification. Also, at present, much of the previous research on HIIT in older populations has included individuals in the 5th decade of life. This makes definitive qualification of “older” impractical. We have adopted your suggestion to an extent, by explicitly stating the mean age of each study introduced and discussed. This can then be compared against our participant data for more granular comparisons. Please refer to the paragraph starting at line 43 for how we did this

Point 2: On page 2, line 49-50, when discussing the Herbert study, please explain how "delayed recovery" was defined or quantified.

Response 2: The Herbert study quantified recovery as the ability to produce peak power output, we have now stated this in line 53 with the following text:
However, Herbert et al. [21] observed a delayed recovery of PPO in older participants with a mean age of 63 years.

Point 3: Lines 78-81: Why were 3 and 5 days chosen? To my knowledge, most studies utilizing HIIT/SIT as a training methodology use ~48 hours or 2 days between training sessions. A stronger rationale is needed here.

Response 3: A delayed recovery has been previously observed after 3 days of rest with HIIT in older adults [21]. Therefore, a shorter recovery duration was not in the scope of this study, given previous evidence. Given that age is a key component of muscle function and recovery, discussing data from typically young, athletic cohorts may serve to confuse the discussion somewhat. Other evidence, such as those discussed in reference 11 of our paper (48 hours or 2 days as you suggested), are conducted in young and athletic populations. As such, they are not suited to designing research with older adults. We have now highlighted this with the following in paper text:
A previous review has discussed literature on HIIT inter-session recovery, concluding optimal recovery to be approximately 48 hours following HIIT [11]. However, the research discussed in this review is of young and athletic populations, who are likely to recover faster due to age related biological factors [22-24]. Therefore, the recovery timeframes utilised by Herbert et al., [21] were adopted.  

Point 4: Methods: Were participants given any pre-testing standardization instructions regarding physical activity and/or diet? These are potentially important confounders that should be addressed.

Response 4: Under section 2.1, we have listed the extent of controls in the study. It was not logistically possible to control for some unsystematic biases. Ideally, we would have been able to control many unsystematic biases. For instance, protein, carbohydrate, fat intake, which all have effects on recovery. Moreover, things like the biological availability of Vitamin D etc could all have an unsystematic bias on the research. However, that was outside the scope of this study. We did however, control for physical activity, alcohol, and caffeine as reported in the methods.

Point 5: Results: While I understand the primary outcome was Peak Power, it would also be interesting to present data for mean power, minimum power, and/or fatigue index if these data are available. I feel this would strengthen the paper's primary outcomes.

Response 5: The Herbert 6 s test is validated for PPO testing, as such, we did not record mean power or fatigue index. The 6 s is not relevant in the context of fatigue index or mean power. However, mean and peak power from each sprint is presented in table 1.

Point 6: Discussion: As with the introduction, I recommend quantifying the ages of other studies' participants so that it eases interpretation for the reader. Obviously we would likely expect a study with an average age of 50 to have different outcomes than a study with an average age of 70.

Response 6: Thank you for the suggestion. In-line with this comment, we have adopted this approach when describing all previous literature to provide context for age-related HIIT discussion.

Point 7: Thank you for the opportunity to review the paper.

Response 7: Thank you for your thoughtful suggestions. We appreciate the suggestion to state the mean age of participants studied previously. Additionally, thank you for highlighting the need to further clarify the use of 3 vs 5 days of rest, we feel the explicit statement that resulted from your suggestion better reflects the reasoning for our research design.

Reviewer 2 Report

Overall very interesting and straight forward investigation. There are a few potential problems listed below.

Intro

Line 36- provide more of a distinction of SIT (like >100% VO2max) typically 30s or less of an all-out effort.

Line 44- “healthogenic” is not a term seen often, perhaps define this term.

Line 68- “improvments” is misspelled.

Only tested 3 and 5 days, maybe the fatigue was not captured?

Methods

Were exercise habits similar between the older and the young individuals? Present this data separately and run an independent t-test to make sure that they aren’t different.

Also, why is just PPO used to determine if they are recovered? This is really only testing if one energy system is recovered? What about a mean power for 20s?

Line 123- Cohen’s d, is normally lowercase. Also for these effect sizes, what were the thresholds used to determine if the effect was small, moderate, or large? Were the commons cutoffs used?

Results

Line 127- the age x time interaction actually has a moderate effect. This likely came from the young going down slightly at 3 days and the old going up at the same timeframe.

Discussion

There is a lot of discussion about ROS; however, the exercise bout is likely done through almost exclusively ATP-PC and anaerobic glycolysis, so the mitochondria is probably only being used to resynthesize phosphocreatine. Also mitochondrial coupling actually gets better with exercise, so this is strange to discuss.

Also, comparing to strength studies isn’t really applicable. There is no eccentric loading in cycling, thus little to no muscle damage is accrued. This may be why no decrements were observed. Does cycling in a SIT protocol increase serum creatine kinase or myoglobin?

Probably the biggest limitation was that more timeframes were not tested. Perhaps individuals could do these sprints every day? This is a question that we don’t really know the answer to, two days, 24h?

Author Response

Reviewer 2 comments, and author responses.

Overall very interesting and straight forward investigation. There are a few potential problems listed below.

Intro

Point 1: Line 36- provide more of a distinction of SIT (like >100% VO2max) typically 30s or less of an all-out effort.

Response 1: We have adjusted the manuscript to mention the typically short duration of SIT. However, in our judgement, using >VO2max as a marker of intensity for SIT will likely cause confusion. Our previous work (https://www.ncbi.nlm.nih.gov/pubmed/29701444; https://www.ncbi.nlm.nih.gov/pubmed/28794164; https://www.ncbi.nlm.nih.gov/pubmed/28701523; https://www.ncbi.nlm.nih.gov/pubmed/28042739; https://www.ncbi.nlm.nih.gov/pubmed/28511954; https://www.ncbi.nlm.nih.gov/pubmed/28178145) has used 40% peak power output, which is 96-150% VO2max, suggesting VO2max is not a good measure of 'all-out' ability (i.e. maximal power). Further, this suggests >VO2max is not sufficient to compare to sprinting, as it is only ~40% the power. Moreover, using a percentage of aerobic capacity as a criterion for an anaerobic measure seems counterintuitive. Therefore, when referring to SIT, using maximal effort is more apt.

Point 2: Line 44- “healthogenic” is not a term seen often, perhaps define this term.

Response 2: This has been changed to: indicative of improved health

Point 3: Line 68- “improvments” is misspelled.

Response 3: This has been corrected as suggested

Point 4: Only tested 3 and 5 days, maybe the fatigue was not captured?

Response 4: We agree that we may not have captured fatigue if it was within the first 2 days. Future research could further reduce the possibilities to 1 and 2 days of recovery. However, our research was interested in the 3 v 5-day interval, as there was previous research which had set a precedent, reference [21] in text.

Methods

Point 5: Were exercise habits similar between the older and the young individuals? Present this data separately and run an independent t-test to make sure that they aren’t different.

Response 5: We have conducted an independent t-test as requested in text:

Additionally, an independent samples t-test was conducted to compare weekly mean habitual physical activity at above moderate intensity between older and young participants. Effect size was determined using Cohen’s d, defined as small 0.1, medium 0.3, and large 0.5

Weekly mean habitual physical activity revealed a large effect between older (417 ± 313 minutes) and young participants (310 ± 65 minutes); (t = (8.69) 1.01, p = 0.342, d = 0.68), equal variances were not assumed (p = 0.14).

Point 6: Also, why is just PPO used to determine if they are recovered? This is really only testing if one energy system is recovered? What about a mean power for 20s?

Response 6: This is an interesting suggestion. We have changed the statements in the paper to clearly state that conclusions are limited to the recovery of PPO. As opposed to the vaguer term of “recovery”. Obviously, the term 'recovery' is a pandora's box of variables. We could have measured IL-6, cortisol, testosterone, myoglobin, CK, EMG, perception of recovery, etc. etc. However, these data are outside the scope of this experiment.

Point 7: Line 123- Cohen’s d, is normally lowercase. Also for these effect sizes, what were the thresholds used to determine if the effect was small, moderate, or large? Were the commons cut-offs used?

Response 7: Thank you for the outlining this. We have now corrected this and have stated the cut-offs for the effect size/s.

For the repeated measures ANOVA:
Partial Eta Squared (­­2p) was used as a measure of main effect, defined as small 0.02, medium 0.13, and large 0.26. Cohen's d was calculated for pairwise comparisons

For the independent samples t-test:
Effect size was determined using Cohen’s d, defined as small 0.1, medium 0.3, and large 0.5

Results

Point 8: Line 127- the age x time interaction actually has a moderate effect. This likely came from the young going down slightly at 3 days and the old going up at the same timeframe.

Response 8: This has been corrected in lines 137-138.

Discussion

Point 9: There is a lot of discussion about ROS; however, the exercise bout is likely done through almost exclusively ATP-PC and anaerobic glycolysis, so the mitochondria is probably only being used to resynthesize phosphocreatine. Also mitochondrial coupling actually gets better with exercise, so this is strange to discuss.

Response 9: Thank you for your suggestion and insight. However, please note that although ATP-PCr and anaerobic glycolysis are the major contributors to output, the oxidative system is still stressed significantly by a 20 s bout of maximal exercise, please refer to Baker et al., (2010) for further information on the metabolic response to intense exercise. This is also the case for a ‘varying between individuals’ portion of the self-paced recovery due to the oxygen debt accumulated during sprints.

The reason for the discussion was in relation to the previous study by Herbert et al., (2015). They used a 6 x 30 s at 50% of peak power output protocol. This would very likely have stressed the oxidative system more than our sprints. Therefore, we have speculated that the ROS generation of their protocol is higher than ours. Regarding altered ROS regulation with training status, we have covered this in the paper with the following excerpt:

“Although reactive oxygen species are facilitative ­­­of physiological adaptations, it is theorized that there is an optimal reactive oxygen species production threshold, influenced by exercise intensity and/or duration, which may be altered with training status [44], and age [24].”

Additionally, generation of ROS are multifaceted, please refer to reference [44] in our text for further information. We have however, attempted to clarify this further with the addition of the following text:

“Furthermore, it is noteworthy to mention that regular exercise has demonstrably improved ROS regulation [44]. Therefore, it is probable that ROS regulation would adapt with exercise habituation.”

It is worth noting that, older adults are likely to have a higher contribution from their oxidative energy system during sprints due to a shift (type 2 to type 1) in muscle fibre type. Also, due to decreased neural function, older individuals are not as capable of producing anaerobic work. This is somewhat countered by the overall decrease in oxidative capacity in older adults. However, decrements to anaerobic work potential are greater. This part of the discussion is required to suggest a future area of research. It would also be a good research question to look for changes to ROS bio-markers during an exercise intervention for future research. Perhaps, this would be useful in identifying the ‘rate of change’ in ROS regulation. Yet, if the reviewer suggests removing this topic, we are happy to do so.

Point 10: Also, comparing to strength studies isn’t really applicable. There is no eccentric loading in cycling, thus little to no muscle damage is accrued. This may be why no decrements were observed. Does cycling in a SIT protocol increase serum creatine kinase or myoglobin?

Response 10: This part of the discussion is required to introduce a broader context of exercise prescription guidelines in older adults. We have adjusted the text slightly to clarify this focus with the following text:

Current physical activity guidelines for older people suggest that at least 150 minutes of moderate, or, 75 minutes of vigorous aerobic exercise should be accumulated weekly in at least 30- or 10-minute bouts respectively [40]. Additionally, Chodzko-Zajko et al. [40], suggested a resistance training frequency of twice per week. Currently however, there is no comparable consensus on HIIT or SIT for older adults.  

We are not aware of any good quality studies that demonstrate CK or myoglobin increases during cycling SIT. In line with your suggestion, we have removed the following statement:

Although neither HIIT or SIT can be clearly categorized as either strength or aerobic training, it may be prudent to assume it is placed in between the two.

Point 11: Probably the biggest limitation was that more timeframes were not tested. Perhaps individuals could do these sprints every day? This is a question that we don’t really know the answer to, two days, 24h?

Response 11: Future research could further reduce the possibilities to 1 and 2 days of recovery. However, our research was interested in the 3 v 5-day interval, as there was previous research to compare to. Our future work will attempt to answer this question. However, it is worth noting that, previous work in athletic populations has concluded optimal recovery to be ~ 48 hours, with updated text cited in the introduction:

A previous review has discussed literature on HIIT inter-session recovery, concluding optimal recovery to be approximately 48 hours following HIIT [11]. However, research discussed in this review concerned young and athletic populations, who likely recover faster due to age related biological factors [22-24]. Therefore, the present study examined recovery timeframes utilised by Herbert et al. [21].